



# From emission scenarios to spatially resolved projections with a chain of computationally efficient emulators: MAGICC (v7.5.1) – MESMER (v0.8.1) coupling

Lea Beusch[1], Zebedee Nicholls[2,3,4], Lukas Gudmundsson[1], Mathias Hauser[1], Malte Meinshausen[2,3,4], and Sonia I. Seneviratne[1]

[1]Institute for Atmospheric and Climate Science, ETH Zurich, Zurich, Switzerland
[2]Climate and Energy College, The University of Melbourne, Parkville, Victoria, Australia
[3]School of Geography, Earth and Atmospheric Sciences, The University of Melbourne, Parkville, Victoria, Australia
[4]Climate Resource, Northcote, Victoria, Australia

**Correspondence:** Lea Beusch (lea.beusch@env.ethz.ch)

**Abstract.** Producing targeted climate information at the local scale, including major sources of climate change projection uncertainty for diverse emissions scenarios, is essential to support climate change mitigation and adaptation efforts. Here, we present the first chain of computationally efficient Earth System Model (ESM) emulators allowing to rapidly translate greenhouse gas emission pathways into spatially resolved annual-mean temperature anomaly field time series, accounting for

both forced climate response and natural variability uncertainty at the local scale. By combining the global-mean, emissions-driven emulator MAGICC with the spatially resolved emulator MESMER, ESM-specific as well as constrained probabilistic emulated ensembles can be derived. This emulation chain can hence build on and extend large multi-ESM ensembles such as the ones produced within the 6[th] phase of the Coupled Model Intercomparison Project (CMIP6). The main extensions are threefold. (i) A more thorough sampling of the forced climate response and the natural variability uncertainty is possible with millions

of emulated realizations being readily created. (ii) The same uncertainty space can be sampled for any emission pathway, which is not the case in CMIP6, where some of the most societally relevant strong mitigation scenarios have been run by only a small number of ESMs. (iii) Other lines of evidence to constrain future projections, including observational constraints, can be introduced, which helps to refine projected future ranges beyond the multi-ESM ensemble's estimates. In addition to presenting results from the coupled MAGICC-MESMER emulator chain, we carry out an extensive validation of MESMER,

which is trained on and applied to multiple emission pathways for the first time in this study. The newly developed MAGICC-MESMER coupled emulator will allow unprecedented assessments of the implications of manifold emissions pathways at regional scale.

## 1   Introduction

Earth System Models (ESMs) are the primary tools to study the impact of greenhouse gas emissions on our climate (IPCC,

2013). While the insights they provide are invaluable to advance our understanding of the coupled Earth System to external influences, their projections are affected by three major sources of uncertainty: (i) internal variability, i.e., unforced natural





climate variability; (ii) forced climate response uncertainty, i.e., uncertainty in the response of the climate system to both forced natural (solar and volcanic) and anthropogenic (greenhouse gases, aerosols, land-use change, etc.) influences; and (iii) emission scenario uncertainty, i.e., which emission pathway the world chooses (Hawkins and Sutton, 2009; Lehner et al.,

2020). Each of these uncertainty classes again encompass a myriad of different contributions to the total uncertainty, e.g., carbon cycle uncertainty, aerosol forcing uncertainty, and climate sensitivity uncertainty are all captured within the climate response uncertainty in the above categorization. Due to their high computational cost, ESMs can only sparsely explore the full uncertainty phase space.

This sparse exploration is problematic because targeted climate information accounting for all major sources of climate

change uncertainty is urgently needed, especially given that both Earth's climate (IPCC, 2013, 2018) and the future emission pathway the world's nations have pledged to follow (CAT, 2019, 2021) are changing rapidly. When assessing the implications of a large number of emission pathways for future climate, it is neither computationally feasible nor efficient to create full ESM ensembles for each emission pathway, especially ESM ensembles which thoroughly sample the natural variability and climate response uncertainty space. Instead, computationally efficient ESM emulators could be useful tools to provide targeted climate

information for a few key variables, such as surface air temperatures.

Thus far, ESM emulators have primarily been employed to swiftly translate emission pathways into global-mean climate projections, most prominently forced global temperature change (Meinshausen et al., 2009; Clarke et al., 2014; Rogelj et al., 2016, 2018; Nicholls et al., 2021c; CAT, 2021). Regional climate information would constitute a valuable addition, since it is more directly related to climate change impacts and the climate that people experience (Seneviratne et al., 2016). Spatially

resolved climate information accounting for both climate response uncertainty and natural climate variability for a given emission pathway would be especially useful for policy makers to understand the implications of mitigation efforts for their own country and to assess what climate change adaptation measures need to be implemented.

Here, we present the first chain of computationally efficient ESM emulators able to translate user-defined emission or concentration scenarios into spatially and temporally resolved temperature anomalies with respect to a pre-industrial baseline

accounting for all major sources of climate change uncertainty (Fig. 1). The global MAGICC emulator (Meinshausen et al., 2009, 2011) is used to turn greenhouse gas emission – or concentration – pathways into constrained forced global temperature change time series by taking multiple lines of evidence into account. These are then translated into temperature anomaly field time series, accounting for both regional forced climate response and internal natural variability uncertainty, with the spatially resolved MESMER emulator (Beusch et al., 2020a), calibrated on a range of ESMs from the 6[th] phase of the Coupled Model

Intercomparison Project (CMIP6; Eyring et al., 2016) ensemble.

Figure 2 illustrates how the sources of uncertainty accumulate at the local scale by presenting time series for a single grid point in Eastern North-America as well as maps of example realizations in 2030 for a low emission scenario. In the most basic setup of the MAGICC-MESMER chain, a single forced global mean temperature change time series can be coupled to a single set of local mean response parameters, resulting in a single realization (Fig 2a). When accounting for global climate response

uncertainty by using MAGICC's probabilistic distribution of forced global mean temperature time series in combination with the single local mean response parameter set, different realizations are obtained but each emulation ensemble member exhibits

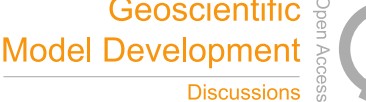

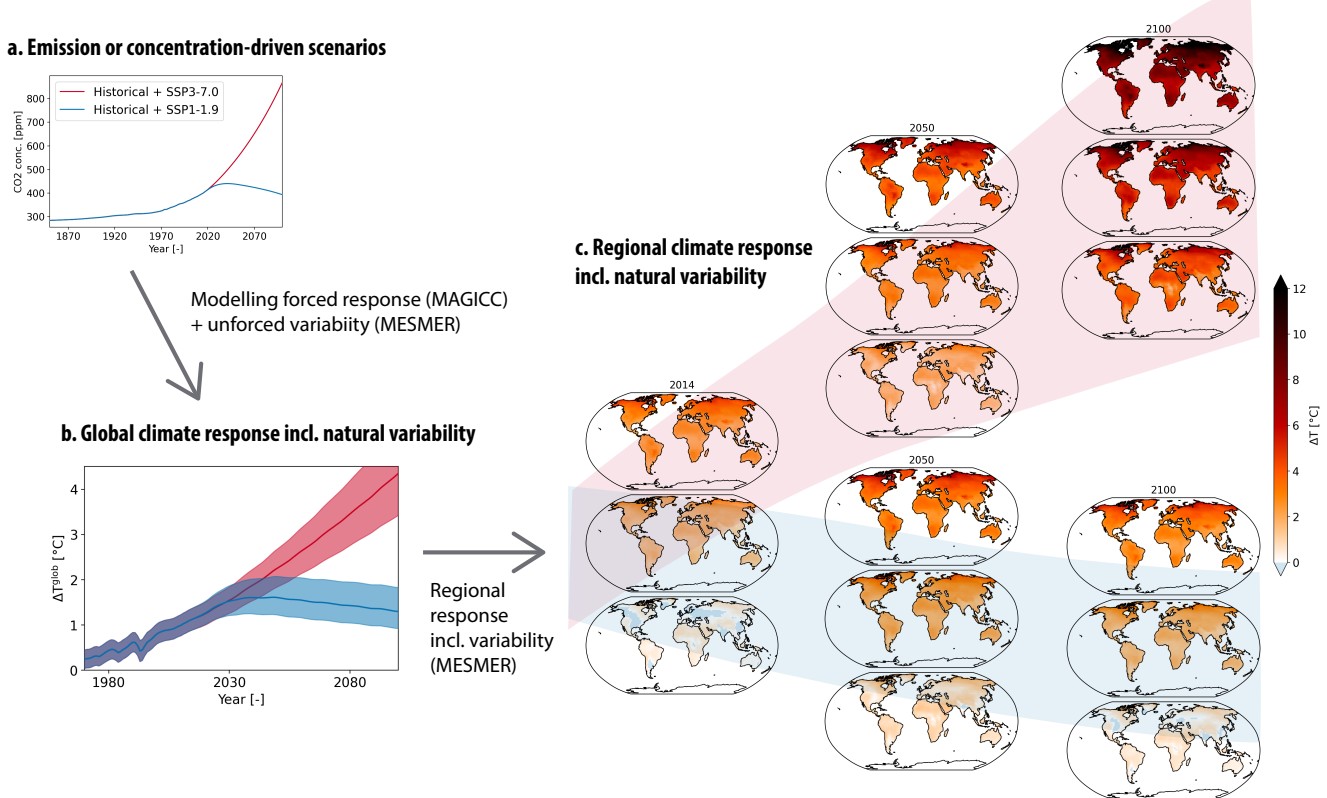

**a. Emission or concentration-driven scenarios**

Modelling forced response (MAGICC)
+ unforced variabiity (MESMER)

**b. Global climate response incl. natural variability**

Regional
response
incl. variability
(MESMER)

**c. Regional climate response
incl. natural variability**

**Figure 1.** Illustration of the MAGICC-MESMER emulator chain. (a) The sequence of the analysis starts from either emission or concentration pathways, as highlighted here for the $CO_2$ concentration time series of the historical time period and two Shared Socioeconomic Pathway (SSP) scenarios (O'Neill et al., 2017), namely SSP1-1.9 and SSP3-7.0. (b) The probabilistic global mean temperature change distributions – whose medians and 90 % ranges (5th – 95th percentile) are shown – consist of realizations which are a combination of the scenario-specific forced global temperature change from MAGICC and emulated natural variability from MESMER. (c) Based on this information, MESMER can derive the associated spatially resolved temperature change distributions, with the maps shown here representing the 5th percentile, the median, and the 95th percentile (each map trio from bottom to top) for 2014 (first column of maps), for 2050 for both SSP scenarios (second column of maps), for and 2100 for both SSP scenarios (third column of maps).

the same spatial pattern (Fig 2b). Different spatial patterns become available once the uncertainty in the regional forced response is included and the forced global temperature time series are combined with each of the different available ESM-specific local forced response parameter sets (Fig. 2c). The last source of uncertainty is ESM-specific natural climate variability, which is added on top of the local forced response patterns (Fig. 2d). In this low emission scenario, natural variability accounts for roughly half of the uncertainty at the local scale even at the end of the 21st century. If a high emission scenario were studied instead, the overall spread at the end of the century in terms of regional forced response would increase, and thus, natural variability would be less important compared to the other sources of uncertainty.


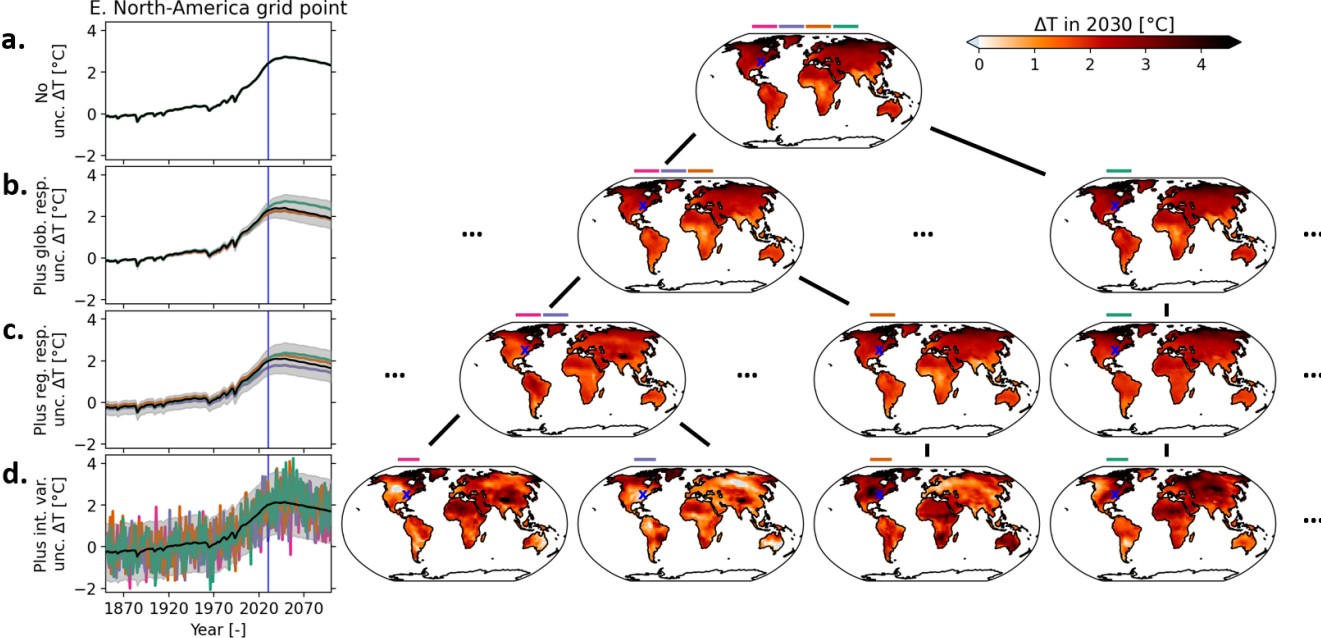

**Figure 2.** Visualization of the cumulative contribution of different uncertainty sources accounted for in the MAGICC-MESMER emulator chain for a single grid point in Eastern North-America and example realizations on maps in 2030 (indicated in blue) for the low emission scenario SSP1-1.9. The emulations which are shown in the maps are also depicted in the grid-point-level time series by using the same color coding. From top to bottom, the rows exhibit an increasing reflection of uncertainties: (a) Accounts for no uncertainty since a single MAGICC time series is combined with a single MESMER forced response pattern. (b) Accounts for the uncertainty in the global response by combining all MAGICC time series with a single MESMER forced response pattern. In the time series plot, the solid black line indicates the median of the temperature change distribution and the gray shading the 90 % range (5th – 95th percentile). (c) Adds uncertainty in the regional response by combining all MAGICC time series with all of MESMER's forced response patterns. (d) Adds natural internal variability derived by MESMER on top of the forced response patterns.

While this study is, to our knowledge, the first to combine all of these uncertainties in a single emulator chain, rich back-
ground literature exists on each individual uncertainty aspect. The uncertainty in the global climate response to greenhouse gas emissions is modeled by a variety of global emulators, which usually have a physical core and use different statistical approaches during calibration (Nicholls et al., 2021c). The uncertainty in the local forced response to global temperature is most frequently accounted for through different flavours of pattern scaling (Tebaldi and Arblaster, 2014; Lynch et al., 2017; Beusch et al., 2020a). The last element of our emulation chain, local-scale natural variability, has been stochastically created through
re-sampling of either individual temperature field realizations (Alexeeff et al., 2018) or of principal components with perturbed phases (Link et al., 2019), and by sampling from autoregressive processes with spatially correlated innovation terms (Beusch et al., 2020a). Additionally, there are two studies which directly translate emissions into spatially resolved temperature change realizations (Goodwin et al., 2020; Yuan et al., 2021). However, both of the available approaches lack one major source of





uncertainty. Goodwin et al. (2020) combine the global emulator WASP with a pattern scaling approach to obtain probabilistic
mean warming realizations but do not account for internal variability. Yuan et al. (2021), on the other hand, use a statistical
emulator calibrated on a single ESM to create spatially resolved temperature realizations directly from $CO_2$ equivalence con-
centrations and thus neither account for forced climate response uncertainty nor uncertainty in the representation of unforced
natural climate variability.

## 2   Data

### 2.1   Earth System Model data

We use climate information from 25 ESMs participating in the Scenario Model Intercomparison Project (ScenarioMIP; O'Neill
et al., 2016) of CMIP6 (Eyring et al., 2016) listed in Appendix Table A1. While our ESM ensemble does not contain all of
the ESMs available within CMIP6, it does cover a wide range of temperature realizations and is broadly representative of the
overall warming spread and the relative fraction of simulations available for each scenario.

Data from $1850 - 2100$ are employed, covering the historical time period ($1850 - 2014$) and the ScenarioMIP's future ($2015 -$
$2100$) emission and land use change scenarios, the Shared Socioeconomic Pathways (SSPs), i.e., SSP1-1.9, SSP1-2.6, SSP4-
3.4, SSP5-3.4-over, SSP2-4.5, SSP4-6.0, SSP3-7.0, SSP5-8.5. In this study, a special focus is set on SSP1-1.9, the most ambi-
tious mitigation scenario of CMIP6, and SSP3-7.0, a physically plausible high-end emission scenario (Hausfather and Peters,
2020).

We primarily employ annual 2-m air temperature ($T$) data but ocean heat uptake ($OHU$) is additionally used for some
analyses. Both variables are re-gridded onto a common $2.5° \times 2.5°$ spatial grid according to Brunner et al. (2020a). Anomalies
with respect to $1850 - 1900$ are obtained at the grid point level and used for all analyses.

Regional averages refer to area-weighted averages and regional land averages to area- and land-fraction-weighted averages.
Grid cells with more than one third land fraction are considered to be land grid points. The global land average does not
include Antarctica, since no emulations are created for Antarctica. The Eastern North-America region frequently employed in
this study is one of the updated Intergovernmental Panel on Climate Change (IPCC) climate reference regions (Iturbide et al.,
2020) and is extracted from the gridded fields with the help of the regionmask package (Hauser et al., 2020).

### 2.2   Observational data

For a qualitative visual validation, annual blended temperature anomalies from the Berkeley Earth data set are employed (Rohde
et al., 2013), which consist of land and sea ice 2-m air temperature anomalies and ocean surface temperature anomalies. They
are interpolated onto the same spatial grid as the ESM data, also following the approach described in Brunner et al. (2020a).
To account for differences in the global average of blended temperature anomalies and and 2-m air temperature anomalies, the
observational global blended averages $\Delta T^{glob,blend}$ are transformed into 2-m air global averages $\Delta T^{glob,air}$ at every point in
time $t$ via the following formula $\Delta T_t^{glob,air} = 1.098 \cdot \Delta T_t^{glob,blend} - 0.001°C$ derived by Beusch et al. (2020b) based on CMIP6





data. Additionally, the observational data needs to be shifted from their native $1951-1980$ baseline to the $1850-1900$ baseline employed within this study. However, during the $1850-1900$ time period, the observational data are not spatially complete and the quality of the available data is lower. Hence, the MAGICC-MESMER emulated regional median warming between the $1850-1900$ baseline and the native baseline of the observational data is added to the observational data as a constant offset during plotting.

## 3   Methods

### 3.1   MAGICC (v7.5.1)

The Model for the Assessment of Greenhouse gas Induced Climate Change (MAGICC) is a reduced complexity climate model, which calculates – among other quantities – forced global warming and global ocean heat uptake. Its hemispherically-resolved multi-layer upwelling-entrainment-diffusion ocean and climate core (based on the energy balance equation with state-
and forcing-dependent climate sensitivity) are described in Meinshausen et al. (2011) and Meinshausen et al. (2020). It also includes representations of the carbon cycle, non-CO2 greenhouse gas cycles and the relationship between aerosol prescursor species emissions, and aerosol effective radiative forcing (Meinshausen et al., 2011, 2020) alongside a parameterization of the response of permafrost to global heating (Schneider von Deimling et al., 2012).

The MAGICC output employed throughout this paper is the same as presented in Nicholls et al. (2021c). We use greenhouse
gas concentration-driven simulations, following the CMIP6 ScenarioMIP approach (O'Neill et al., 2016). A slightly wider output temperature distribution would be achieved, if MAGICC were run in emission-driven mode instead (Nicholls et al., 2021c).

### 3.2   MESMER (v0.8.1)

#### 3.2.1   Default configuration – Local temperature anomalies as a function of global temperature anomalies

MESMER, a Modular Earth System Model Emulator with spatially Resolved output, was introduced by Beusch et al. (2020a) to emulate multi-ESM initial-condition ensembles for a specific emission pathway. This first MESMER variant has been successfully applied to emulate temperature changes for the highest emission scenarios of both the CMIP5 (Beusch et al., 2020a) and the CMIP6 (Beusch et al., 2020b) ensembles.

MESMER is an ESM-specific emulator which needs to be calibrated for each ESM it emulates separately, to capture the
unique characteristics of that ESM, both in terms of local forced warming and local variability around the forced warming (Beusch et al., 2020a). Within MESMER, local temperature anomalies $\Delta T$ for a specific climate model $m$ at every point in space $s$ and time $t$ are emulated as follows:

$$\Delta T_{m,s,t} = f(\Delta T_{m,t}^{glob}) + \eta_{m,s,t} = \beta_{m,s}^{fr} \cdot \Delta T_{m,t}^{glob,fr} + \beta_{m,s}^{iv} \cdot \Delta T_{m,t}^{glob,iv} + \beta_{m,s}^{intercept} + \eta_{m,s,t}. \tag{1}$$





The local temperature anomaly is a direct function of global mean temperature change $\Delta T^{glob}$ and a spatio-temporally cor-
related noise term $\eta$. At every point in space, a multiple linear regression is employed to relate $\Delta T^{glob}$ information to local tem-
perature anomalies. Forced global temperature change $\Delta T^{glob,fr}$ and internal global temperature variability $\Delta T^{glob,iv}$ serve
as predictors. The associated regression coefficients are $\beta^{fr}$ and $\beta^{iv}$, and $\beta^{intercept}$ constitutes the intercept term. $\Delta T^{glob,fr}$
is obtained by first applying locally weighted scatterplot smoothing (LOWESS) to the $\Delta T^{glob}$ time series and subsequently
adding volcanic spikes to the smooth forced temperature time series via a linear regression of the global temperature anomaly
residuals to stratospheric aerosol optical depth. The remaining internal global variability $\Delta T^{glob,iv}$ and the residual local vari-
ability $\eta$ are both modeled as autoregressive processes. For the $\eta$, spatially correlated innovation terms are drawn to account
for local to regional cross-correlations. The full algorithm is described in Beusch et al. (2020a).

In this work, MESMER is for the first time trained on and applied to the full range of emission scenarios covered within the
CMIP6 ScenarioMIP projections instead of a single emission pathway. The main additional assumption when extending MES-
MER from emulating a single emission pathway to emulating a range of scenarios is that, once calibrated, the ESM-specific
MESMER parameters can be used to emulate any emission scenario. This assumption of universal scenario applicability of
the calibrated parameters is also regularly made in the well established pattern scaling literature (Tebaldi and Arblaster, 2014),
although care is nonetheless required, particularly for strong mitigation scenarios (Goodwin et al., 2020).

To obtain robust MESMER parameter estimates for each ESM, MESMER is trained on all available ensemble members
of each available scenario and equal weight is given to each scenario. The scenarios employed for training are – subject to
availability – SSP5-8.5, SSP3-7.0, SSP4-6.0, SSP2-4.5, SSP5-3.4-over, SSP4-3.4, SSP1-2.6, SSP1-1.9, and Historical. Note
that we consider the historical time period as its own scenario during the MESMER training. Since different future SSP
projections usually branch from the same historical members, the historical years would receive more weight if the historical
and the future time period were concatenated during training.

Many ESMs have a different number of ensemble members ($n_e$) for each scenario, and the scenarios (historical and projec-
tions) have a different number of time steps ($n_t$), resulting in a different number of samples ($n_s = n_e \cdot n_t$) per scenario.

Therefore, when estimating the coefficients of the multiple linear regression, all simulations from all scenarios are pooled
and each sample (e.g., the temperature anomaly for a specific location, year, and scenario) is weighted by $1/n_s$, i.e., by the
inverse number of samples available for the respective scenario. The same approach is applied when estimating the empirical
spatial covariance of the residual local variability.

For the autoregressive processes, the coefficients are determined for each ensemble member individually and are subse-
quently averaged for each scenario before averaging across all scenarios. For the autoregressive process describing the global
variability, the order of the process must additionally be selected before the parameters are fit. The order is first chosen for each
ensemble member individually based on the Bayesian Information Criterion and then, the median order is identified for each
scenario. Lastly, the median order over all scenarios is selected to describe the global variability of the ESM at hand.





### 3.2.2 Additional predictors configuration – Local temperature anomalies as a function of global temperature anomalies and global ocean heat uptake

To account for potential non-linearities in the climate system (Mitchell, 2003) and for changing forced response warming patterns when moving from a transient to an equilibrium climate (King et al., 2020), we introduce two additional predictors into the

grid-point-level multiple linear regressions of MESMER (Eq. 1). (i) Squared forced global temperature change $(\Delta T^{glob,fr})^2$ is used to represent non-linear feedbacks and (ii) the forced trend in global ocean heat uptake $\Delta OHU^{glob,fr}$ as a proxy for how close the climate system is to equilibrium. $\Delta OHU^{glob,fr}$ is obtained with the same LOWESS approach used to derive $\Delta T^{glob,fr}$ but no volcanic spikes are added, since they are already accounted for in the global temperature response. Hence, the grid-point-level temperature anomalies are emulated as follows:

$$\Delta T_{m,s,t} = \beta_{m,s}^{fr} \cdot \Delta T_{m,t}^{glob,fr} + \beta_{m,s}^{fr2} \cdot (\Delta T_{m,t}^{glob,fr})^2 + \beta_{m,s}^{fr,ohu} \cdot \Delta OHU_{m,t}^{glob,fr} + \beta_{m,s}^{iv} \cdot \Delta T_{m,t}^{glob,iv} + \beta_{m,s}^{intercept} + \eta_{m,s,t}, \qquad (2)$$

where $\beta^{fr2}$ and $\beta^{fr,ohu}$ are the newly introduced regression coefficients.

### 3.2.3 Evaluating both MESMER configurations

When analyzing climate information, it is often helpful to distinguish between forced response and natural variability. In this section, we evaluate both MESMER configurations in terms of their performance in emulating single ESMs with respect to

these two quantities. The two can be separated in a straightforward fashion within the additive MESMER framework (Eq. 1).

The forced local temperature anomaly $\Delta T^{fr}$ of the default MESMER configuration consists of the following terms:

$$\Delta T_{m,s,t}^{fr} = \beta_{m,s}^{fr} \cdot \Delta T_{m,t}^{glob,fr} + \beta_{m,s}^{intercept}. \qquad (3)$$

In the additional predictor configuration of MESMER, the contribution of the additional predictors enters the forced local temperature anomaly too, resulting in

$$\Delta T_{m,s,t}^{fr} = \beta_{m,s}^{fr} \cdot \Delta T_{m,t}^{glob,fr} + \beta_{m,s}^{fr2} \cdot (\Delta T_{m,t}^{glob,fr})^2 + \beta_{m,s}^{fr,ohu} \cdot \Delta OHU_{m,t}^{glob,fr} + \beta_{m,s}^{intercept}. \qquad (4)$$

Figure 3 visualizes the latitudinally-averaged mean local forced warming over the last 30 years for each individual ESM and scenario for both MESMER configurations. For a given scenario, the ESMs' latitudinal patterns exhibit considerable differences. For example, the magnitude of the Arctic amplification (Serreze and Barry, 2011) differs starkly between the ESMs. These regional differences are additionally visualized in maps for select high and low emission scenarios for each ESM in

the supplementary material (Figs. S1 – S5). Overall, the patterns of forced warming are very similar in the default and additional predictor configurations (Fig. 3) and inter-ESM differences are considerably larger than inter-MESMER-configuration differences.

Nevertheless, small but consistent improvements are achieved in most ESMs with the additional predictor configuration (Fig. 3). The magnitude of the improvements depends on the ESM, the scenario, and the latitude. The improvements mostly

occur in the highest emission scenario SSP5-8.5 and in the strong mitigation scenarios SSP5-3.4-over, SSP1-2.6, and SSP1-



**Figure 3.** Latitudinally-averaged grid-point-level emulated local forced warming and performance with respect to ESM simulations for both MESMER configurations averaged over the last 30 years of each scenario. The mean absolute deviation of the ESM simulations from the emulated forced warming, i.e., error, shown here represents the average absolute deviation across all available ESM initial-condition members for the scenario at hand.

1.9. This is an expected consequence of the additional predictors targeting non-linearities and the transition to an equilibrium climate.





Overall, the performance is excellent for both MESMER configurations (Fig. 3): the latitudinal absolute error with respect to the local warming signal rarely reaches 10 % in the future projections. The SSP1 scenarios have a tendency for higher

relative errors, because of their lower overall forced warming signal. This is amplified in the historical period because its low regional warming leads to small absolute errors translating into large relative errors. In the supplementary material (Figs. S1 – S5), average error maps are found to further highlight regional differences in the errors for selected high and low emission scenarios.

Sensitivity experiments are additionally carried out to quantify the impact on performance when using a reduced number

of scenarios during training. For MESMER's default configuration, a generally very similar although slightly reduced performance is achieved by only training on a single high (SSP5-8.5) and low (SSP1-2.6) emission scenario (Fig. S6). If only a single future scenario and the historical time period are used for training, a high emission future scenario results in a better overall performance. This is because training on a low emission scenario requires MESMER to extrapolate to warm climates when emulating higher emission scenarios. Overall, the SSP5-8.5 trained default MESMER configuration often performs compara-

bly to MESMER trained on all scenarios but experiences minor performance reductions in strong mitigation scenarios whose climate moves towards equilibrium conditions (King et al., 2020). The SSP1-2.6 trained emulator, on the other hand, struggles to emulate high emission scenarios (Fig. S6). Hence, for MESMER's default configuration even a single (high emission) scenario largely suffices to emulate a wide range of different emission pathways as long as their emissions do not exceed the ones used for training. The best emulation performance across all scenarios is, however, achieved if both high and low emission

scenarios are included in the training.

If the additional predictors configuration of MESMER is applied instead, it is more important that at least a high and a low emission scenario are available during training because training on a single emission pathway leads to overfitting on the scenario type and thus poor emulation skill in other scenarios (Fig. S7). This mainly occurs because the cross-correlations between the three predictors ($T_{m,t}^{glob,fr}$, $(\Delta T_{m,t}^{glob,fr})^2$, and $\Delta OHU_{m,t}^{glob,fr}$) are fundamentally different in the high emission

transient climate change scenarios compared to the low emission equilibrium-approaching scenarios.

After considering the forced response in detail, we now turn our attention to the local temperature variability when training MESMER on all available scenarios. The ESM-specific emulated local temperature variability around the forced warming consists of the combination between the local response to the global variability and the residual local variability for both MESMER configurations:

$$\Delta T_{m,s,t}^{iv} = \beta_{m,s}^{iv} \cdot \Delta T_{m,t}^{glob,iv} + \eta_{m,s,t}. \tag{5}$$

Figure 4 shows the latitudinally-averaged standard deviation of local variability, analogously to Fig. 3 for each ESM and scenario available for that ESM. However, the standard deviations are computed over the full scenario length instead of only over the last 30 years to obtain more robust estimates. Additionally, the standard deviation of the emulated variability is identical for every scenario since no scenario dependence is integrated in the variability emulation. Differences in the error

between different scenarios for individual ESMs hence solely occur because the ESMs' variability differs from simulation to simulation.

**Figure 4.** Latitudinally-averaged grid-point-level standard deviation of emulated local variability and performance with respect to ESM simulations' standard deviations for both MESMER configurations averaged over the full scenario period. The standard deviation of the local variability emulations is based on 600 emulations for each ESM. To obtain local variability from ESM simulations, the emulated local forced warming is subtracted from every ESM simulation. Subsequently, the standard deviation of this estimate of an ESM simulation's local variability is computed at every grid point. The mean absolute deviation of the ESM simulations' standard deviations from the emulations' standard deviation, i.e., error, shown here represents the average absolute deviation across all available ESM initial-condition members for the scenario at hand.





Generally, the emulated variability is smallest in the tropics and largest in the northern high latitudes, but considerable inter-ESM differences in the magnitude and spatial distribution exist (Fig. 4). In supplementary material Figs. S8 – S12, spatially resolved maps of the standard deviation of the emulated local variability are shown, which further highlight region-specific characteristics.

Inter-MESMER-configuration differences are only minimal, also in terms of improvements with respect to capturing the characteristics of the local variability of the true ESM simulations (Fig. 4). However, the spatially resolved maps can help pinpoint differences in the errors of the two configurations in some regions (Figs. S8 – S12).

The relative errors in the emulated local variability (Fig. 4) are generally a bit larger than the forced local warming ones (Fig. 3). In part, the larger errors are caused by the fact that percentage differences are considered and the standard deviation of local variability is mostly a small quantity. This impression is reinforced by the fact that the largest differences occur most often in the low variability tropics. A possible physical explanation of the remaining errors is revealed by consulting maps of the average error in the standard deviation (Figs. S8 – S12). For certain ESMs and regions, the standard deviations are overestimated in the high-end scenario but underestimated in the low-end scenario (or the other way round), indicative of changes in local variability across climate states (Olonscheck and Notz, 2017), which are not accounted for in MESMER. In line with this finding, the highest deviations are generally observed for SSP5-8.5 which experiences the most extreme climatic change and thus also the largest changes in variability (Fig. 4). Given the physical implausibility of this high emission scenario (Hausfather and Peters, 2020), one could consider excluding it from the local variability training to further improve the local variability representation in the other scenarios. However, since the absolute differences in standard deviations are nevertheless rather small, we continue to use all scenarios for training.

This detailed MESMER evaluation reveals that the additional predictors do bring subtle but systematic improvements. Nonetheless, the main forced warming signal can already be successfully extracted based on forced global temperature change alone and local variability is generally emulated similarly well in both MESMER configurations. In the following, we thus use the default MESMER configuration instead of the additional predictors one. This minimizes the risk of poorly calibrated parameters when only a limited number of scenarios is available for training in which different predictors are strongly correlated.

## 3.3   Coupling MAGICC and MESMER

MAGICC and MESMER are calibrated individually before using them jointly to create ensembles of spatially resolved emulations. In the coupled MAGICC-MESER emulation mode, MESMER's own statistical estimates of $\Delta T^{glob,fr}$ in Eq. 1 are replaced with MAGICC's $\Delta T^{glob,fr}$ estimates (Sect. 3.2.1), making it possible to also provide spatially resolved emulations for emission scenarios which were not available during the MESMER training.

### 3.3.1   ESM-specific emulations

When emulating a specific ESM with MAGICC, a single global forced temperature anomaly $\Delta T^{glob,fr}$ time series is obtained for every emission scenario. Here, we employ ESM-specific MAGICC output published as part of the Reduced Complexity Model Intercomparison Project (RCMIP) Phase 1 (Nicholls et al., 2020) for two ESMs, CanESM5 and CNRM-CM6-1. Note





that this output was generated with MAGICC v7.1.0.-beta but that very similar results would be obtained with MAGICC v7.5.1, which is employed throughout all other parts of our study.

To obtain full global realizations $\Delta T^{glob}$ for these ESMs, the stochastically-generated ESM-specific global variability of MESMER $\Delta T^{glob,iv}$ is added to MAGICC's ESM-specific $\Delta T^{glob,fr}$ time series. For this study, 600 $\Delta T^{glob,iv}$ emulations are created with MESMER for each ESM. Hence, the MAGICC-MESMER $\Delta T^{glob}$ ensemble also contains 600 realizations per

ESM.

The spatially resolved forced warming fields $\Delta T^{fr}$ are obtained by combining MAGICC's $\Delta T^{glob,fr}$ time series with the associated ESM's local forced response parameters provided by MESMER (Eq. 3). The resulting $\Delta T^{fr}$ field time series is combined with 600 local variability emulations $\Delta T^{iv}$ for that ESM provided by MESMER (Eq. 5), leading to a 600 member ESM-specific MAGICC-MESMER ensemble of $\Delta T$ (Eq. 1).

The ESM-specific MAGICC-MESMER ensembles can be regarded as a direct approximation of very large ESM-initial condition ensembles (Deser et al., 2012, 2020), which can be provided at a negligible computational cost for any emission scenario of interest.

### 3.3.2   Globally-constrained probabilistic emulations

To derive trustworthy probabilistic climate projections, which thoroughly sample the climate response and natural variability

climate change uncertainty space for any emission scenario, observational constraints are often required. At each point of the MAGICC-MESMER emulation chain, an observational constraint could theoretically be introduced.

Several studies employing fundamentally different approaches have all demonstrated that the CMIP6 ESMs which exhibit the strongest forced global future warming are not consistent with observationally-constrained warming estimates (Forster et al., 2019; Beusch et al., 2020b; Brunner et al., 2020b; Tokarska et al., 2020; Ribes et al., 2021; Nicholls et al., 2021c).

In terms of regional response to the global warming, however, most CMIP6 ESMs perform in an observationally-consistent manner in most regions (Beusch et al., 2020b). In line with these findings, the MAGICC-MESMER probabilistic constrained projections are constrained solely at the global level but span the full regional ESM response uncertainty range in this study.

The probabilistic MAGICC output used in this study follows the HadCRUT5 (Morice et al., 2021) calibration of MAGICC presented in Nicholls et al. (2021c) and consists of 600 forced global temperature change $\Delta T^{glob,fr}$ time series per sce-

nario. Beusch et al. (2020b) demonstrated that there is no direct relation between an ESM's performance skill for global-scale response to emissions and regional-scale response to global temperatures. Therefore it is assumed that global- and regional-scale performance are sufficiently decoupled to allow for combining MAGICC's probabilistic output with each of MESMER's ESM-specific local parameter sets. Hence, similarly to the ESM-specific emulation approach, a full initial condition ensemble is created for each of the 600 $\Delta T^{glob,fr}$ time series and each of the 25 ESM-specific parameter sets. For the global temperature

realizations $\Delta T^{glob}$, this means that each of the 600 MAGICC $\Delta T^{glob,fr}$ time series are combined with 600 global variability $\Delta T^{glob,iv}$ realizations of 25 ESMs resulting in a total number of nine million $\Delta T^{glob}$ realizations. For the local temperature anomaly field realizations $\Delta T$, each of MAGICC's $\Delta T^{glob,fr}$ time series is combined with the ESM-specific local parameter sets of MESMER (Eq. 1). This results in 600 local forced temperature change $\Delta T^{fr}$ field time series for each ESM-specific





parameter set (Eq. 3). Each of these local forced temperature field time series is subsequently combined with 600 local vari-
ability $\Delta T^{iv}$ realizations from MESMER (Eq. 5) yielding 600 different initial condition ensembles with 600 members each
(Eq. 1). Thus, 360'000 probabilistic $\Delta T$ realizations are contributed from each set of ESM-specific local parameters and the
full probabilistic ensemble contains nine million emulations.

    The complete-sampling approach we employ here ensures a thorough sampling of the full uncertainty space, but may quickly
lead to computer memory issues. Alternatively, a broad – albeit sparser – sampling of the full climate response and natural
variability uncertainty space could also be achieved by randomly combining single MAGICC time series with single ESM-
specific parameter sets.

## 4    Results

### 4.1    Earth-System-Model-specific temperature projections

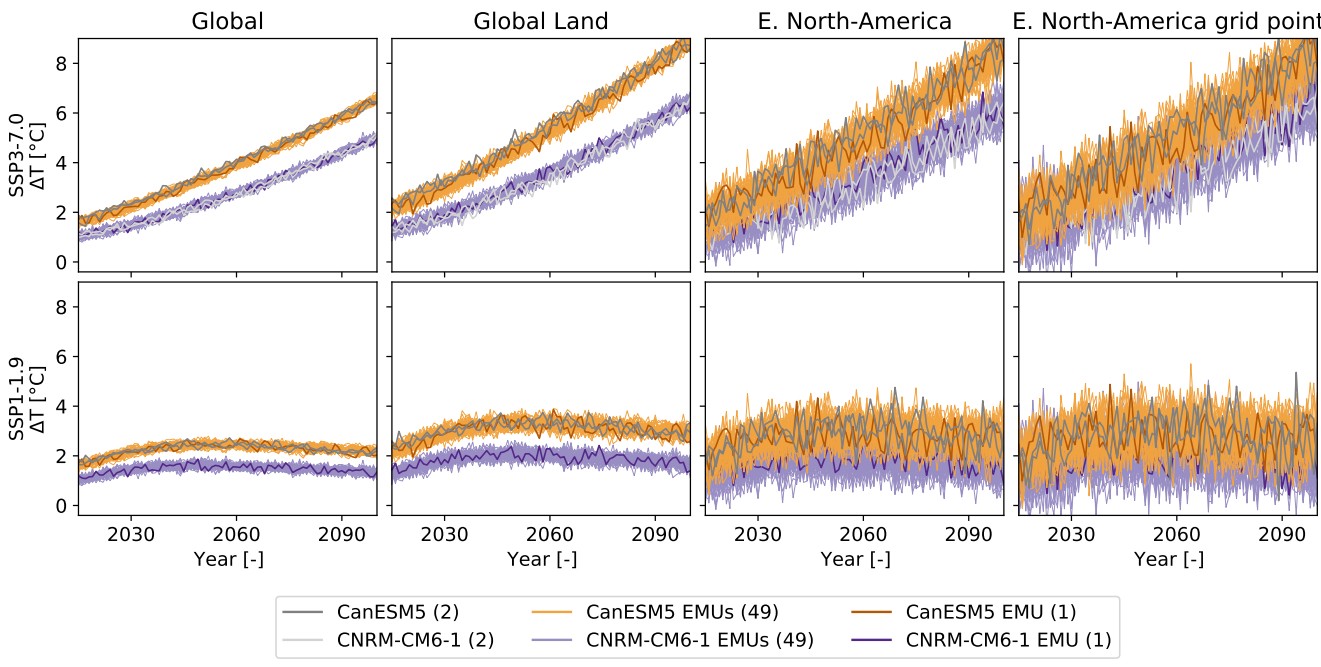

**Figure 5.** Temperature anomaly time series of ESM-specific MAGICC-MESMER emulations (in color) and actual ESM simulations (in
gray), for CanESM5 and CNRM-CM6-1, averaged across different spatial scales: global, global land, Eastern North-America, and a single
grid point within Eastern North-America for a high (SSP3-7.0) and a low emission (SSP1-1.9) scenario. For illustrative purposes, 50 out of
the 600 available emulations are shown for both ESMs in color, and a single emulation is highlighted in darker color. For the ESMs, two
initial-condition ensemble members are shown for each scenario for which simulations are available.





To illustrate the MAGICC-MESMER chain's ability to capture the distinct behavior of individual ESMs in its ESM-specific
configuration, we approximate the behavior of two example ESMs, CanESM5 and CNRM-CM6-1, for two example scenarios,
SSP3-7.0 and SSP1-1.0 (Fig. 5). In both scenarios, the emulations serve to fill the gaps in the natural variability uncertainty
space, by successfully capturing the characteristic ESM-dependent forced warming and climate variability around that warming
for the different spatial scales. The increasing magnitude of natural variability for smaller spatial scales results in an increas-
ing overlap between the two temperature anomaly distributions spanned by the two ESMs. In the low emission scenario,
the emulations additionally fill a gap in the climate response uncertainty space, since no SSP1-1.9 simulations are available
for CNRM-CM6-1. Hence, in this configuration, our emulator approximates the full climate change uncertainty phase space
spanned by the considered ESMs since it can create very large emulated intial-condition ensembles for any given scenario.

It should be noted that the success of these ESM-specific emulations is strongly dependent on MAGICC's ability to match a
given ESM's behavior over a range of scenarios. While we successfully emulate the example ESMs shown here, there are other
ESMs which are less well-captured by MAGICC in this emulation mode (see e.g., Nicholls et al., 2020). On the other hand,
the regional features of single ESMs given the global forced warming trajectories are generally well captured by MESMER
(Sect. 3.2.3). Overall, in the context of ESM-specific emulation, more work on emulating ESMs' forced global temperature
change is required compared to the relative success seen in emulating an ESM's natural variability and regional response to
global warming.

**4.2    Globally-constrained probabilistic temperature projections**

When sampling the full globally-constrained climate response and internal variability uncertainty space with the probabilistic
MAGICC-MESMER ensemble, the emulations no longer coincide with individual ESM simulations (Fig. 6). Instead, the
globally-constrained emulated ensemble encompasses a smaller range of temperature anomalies than the raw CMIP6 ensemble,
and also samples this space much more thoroughly with nine million emulations. This thorough sampling means that even
extreme quantiles at individual grid cells can be statistically robustly estimated for any year. An additional advantage with
respect to the CMIP6 ensemble is that the same forced climate response and natural variability uncertainty space can be
sampled for each scenario, whereas there are stark differences in the number of ESMs providing simulations for each scenario
as well as in the number of simulations of a single ESM for a specific scenario. This is especially relevant, because SSP1-
1.9, which is of great interest for society as it comes closest to the 1.5 °C Paris Agreement target (UNFCCC, 2015), is one
of the scenarios which has only been run by a rather small – and warm – subselection of ESMs. In our CMIP6 ensemble,
only six ESMs provide SSP1-1.9 simulations two of which warm unrealistically fast during the historical period in terms of
global mean and continue to do so in the future (see e.g., Tokarska et al., 2020; Beusch et al., 2020b). The global warming of
these simulations is clearly incompatible with the globally-constrained emulated ensemble (Fig. 6). The two ESMs' global and
global land mean temperature anomalies are almost constantly above the 95th percentile of the emulated ensemble. Also at
the regional and the grid-point level, the constrained ensemble warms distinctly less than these high warming simulations, but
the larger internal variability result in a partial overlap between those ESM's realizations and the climate change uncertainty
(5th – 95th percentile) of the emulated ensemble.



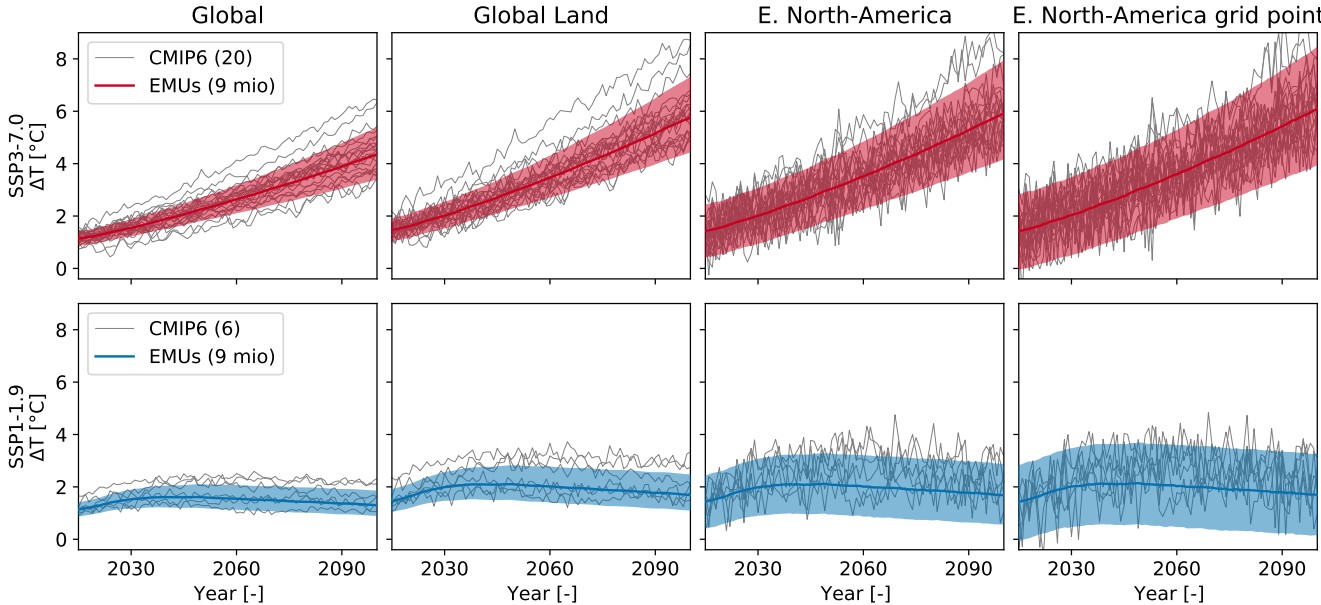

**Figure 6.** Temperature anomaly projection distributions for the emulated globally-constrained probabilistic MAGICC-MESMER ensembles and actual ESM simulations averaged across different spatial scales: global, global land, Eastern North-America, and a single grid point within Eastern North-America for a high (SSP3-7.0) and a low (SSP1-1.9) emission scenario. For the MAGICC-MESMER ensemble, the median and the 90 % range (5th – 95th percentile) of the temperature anomaly distribution are shown in color. For the ESMs, a single simulation per available ESM for that scenario is shown in gray.

To qualitatively validate the probabilistic MAGICC-MESMER ensemble and to highlight further differences to the raw CMIP6 ensemble, we additionally turn back to the time period covered by observations (Fig. 7). For the spatial scales and locations shown here, the MAGICC-MESMER ensemble captures the key characteristics of the observations both in terms of forced warming and scale-specific variability around the forced warming. In line with observations and opposed to CMIP6, the emulated ensemble does not contain any extreme outlier realizations which exhibit a drastic cooling after the 1950s. Hence, the emulated ensemble filters out physically implausible ESM simulations which affect the overall distribution of the CMIP6 ensemble.

## 5 Discusssion: Potential further extensions

### 5.1 Going beyond global mean temperature as a predictor for the regional scale

The careful evaluation in Sect. 3.2.3 showed that even when emulating multiple emission scenarios, a representation of local temperature realizations as a linear function of global forced warming and natural variability is sufficient. Nevertheless, the current MESMER implementation can ingest additional predictors, as highlighted for the squared forced global temperature



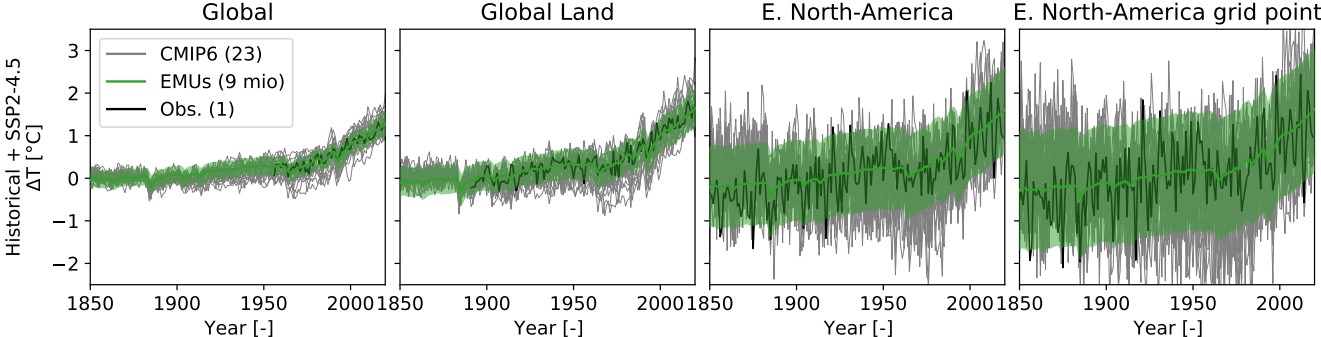

**Figure 7.** Temperature anomaly distribution for the emulated globally-constrained probabilistic MAGICC-MESMER ensemble, actual CMIP6 simulations, and observations averaged across different spatial scales: global, global land, Eastern North-America, and a single grid point within Eastern North-America for the period covered by observations, i.e., the historical time period extended with the middle of the road future scenario (SSP2-4.5). For the MAGICC-MESMER ensemble, the median and the 90 % range (5th – 95th percentile) of the temperature anomaly distribution is shown in color. For the ESMs, a single simulation per available ESM for that scenario is depicted in gray. The observations are shown in black.

and smoothed forced global ocean heat uptake in Sect. 3.2.2. This ability to include multiple predictors is expected to become especially useful once MESMER will be extended to model variables other than annual-mean temperature. For example, annual-mean forced precipitation changes are known to depend on the greenhouse gas and aerosol compositions of the emission pathway, both at global (Ramanathan et al., 2001; Frieler et al., 2012; Pendergrass et al., 2015) and at regional (Frieler et al., 2012) scales. This documented emission pathway dependence could readily be integrated into the MAGICC-MESMER

emulator chain since greenhouse gas and aerosol concentrations are direct outputs of the MAGICC emulator.

However, great care should be taken when deciding to introduce additional predictors into MESMER. The additional predictors should be sufficiently decoupled from the original predictors in the range of emission scenarios used for training to avoid artifacts in the calibrated parameters stemming from cross-correlations in the training data and leading to poor emulation skill in different scenarios. For example, if ocean heat uptake is included as a predictor, it is vital that the training data contains

both a high-end and low-end emission scenario. If MESMER were trained only on a high emission scenario, the calibrated parameters would not be well defined because global temperature and ocean heat uptake are strongly correlated throughout the full scenario. Hence, poor emulation skill would be expected if this MESMER calibration were used in a low emission scenario, in which the two predictors are no longer correlated.

Additionally, when using the full MAGICC-MESMER emulator chain, instead of MESMER alone, it should be verified

that MAGICC successfully emulates each of the additional predictors and that its internal cross-correlations between the predictors are similar to the ones found within the ESMs used to train MESMER. Naturally, deviations from these inter-predictor relationships are acceptable for physically-justified reasons, such as too strong feedbacks within individual ESMs.





However, this would raise the question of whether these ESMs' regional climate response uncertainty should be excluded from the climate change uncertainty space sampled by the MAGICC-MESMER ensemble.

## 5.2 Constraints on regional scale

In this study, we solely constrain the quantity best documented to be in need of a constraint: the global response to changes in greenhouse gas concentrations and anthropogenic aerosol precursor emissions (see Sect. 3.3.2). In Sect. 4.2, we show that the resulting emulated MAGICC-MESMER ensemble exhibits a smaller spread than the raw CMIP6 ensemble, especially reducing the high-end global warming estimates in line with published literature (e.g., Brunner et al., 2020b; Tokarska et al., 2020), and that it can successfully approximate observed warming at various spatial scales.

However, to further improve the regional accuracy of the emulated ensemble's projections, a regional constraint could be applied in addition to the global constraint. Beusch et al. (2020b) propose a first regional-scale observational constraint by identifying ESMs which have a regional response to global warming which can be regarded as consistent with observations and by only including those ESMs' climate response and natural variability uncertainty when deriving regionally-optimized projections. To account for inter-ESM dependencies (Abramowitz et al., 2019; Knutti, 2010), they choose the simplest possible way and only consider one ESM per "ESM name family", e.g., only one of the CNRM ESMs. Their constraint could be further refined to account for inter-ESM dependencies and consistency with observations in a more elaborate way, potentially moving towards a Bayesian constraining framework and introducing Monte-Carlo sampling of parameters like in MAGICC (Meinshausen et al., 2009). Additional thoughts should be put into explicitly constraining natural variability, which differs starkly between different ESMs (Beusch et al., 2020a; Deser et al., 2020) but has thus far received considerably less attention than the forced response.

## 5.3 Exploring regional climate change uncertainty beyond the MAGICC-MESMER coupling

With the MAGICC-MESMER coupling we are able to thoroughly sample the climate response and natural variability uncertainty for arbitrary emission scenarios from global to spatially resolved local scale. However, the MAGICC-MESMER coupling is still confined in its representation of local-scale climate change uncertainty by MAGICC's representation of the global response to greenhouse gas emissions uncertainty and by MESMER's representation of the local response to global climate information uncertainty and its local variability uncertainty. A straightforward way to address this issue, would be to additionally combine different global and regional ESM emulators with each other to ultimately create multi-emulator based probabilistic emulations.

On the global emulator side, first progress towards this direction has already been achieved with the Open Simple Climate Models (OpenSCM) initiative, which aims to bring different global emulators together and to provide standardized output for them. A uniform interface for emissions-driven runs, OpenSCM-Runner (Nicholls et al., 2021b), is available. The implementation for MAGICC is fully functioning and employed throughout this study. FaIR (Smith et al., 2018; Leach et al., 2021) and CICERO-SCM (Skeie et al., 2017, 2021) are two other global emulators which are actively working towards being integrated into the OpenSCM initiative and more global emulators have expressed their interest in joining. Once additional global em-





ulators are available within the OpenSCM framework, assessing the implications of their forced global warming distribution differences for regional scale realizations will be straightforward.

For regional-scale emulators, no such common framework exists to date. However, in addition to MESMER, the fldgen emulator (Link et al., 2019; Snyder et al., 2019) is publicly available and could be coupled to the same global emulators. With
this, the local scale uncertainty introduced by different regional emulator's representation of local-scale climate change could be quantified.

Depending on the scientific question, different emulation strategies are called for. If the aim is to sample as broad an uncertainty space as possible, several global and regional emulator combinations should be included when deriving probabilistic emulated ensembles. On other occasions, it may be more beneficial to try and identify the best performing emulator pair for
the question at hand. After all, each emulators comes with its own sets of assumptions and thus most appropriate domain of applicability.

## 6   Conclusions

In this paper, we present the MAGICC-MESMER emulator chain. To the best of our knowledge, this is the first emulator chain that can be directly used to rapidly assess the implications of different emissions scenarios for temperature changes
at global to local spatial scales while accounting for both forced climate response and natural variability uncertainty. With illustrative examples, it is demonstrated that the MAGICC-MESMER chain is able to successfully provide ESM-specific as well as globally-constrained probabilistic emulations, for a range of emission pathways. While MAGICC is run here in greenhouse gas concentration-driven mode, following the CMIP6 ScenarioMIP approach, MAGICC could alternatively directly translate emissions into the predictors needed by MESMER. The default configuration of MESMER, which is employed for most of our
analyses, only requires forced global temperatures change as a predictor from MAGICC. However, the local forced warming module of MESMER is also able to ingest additional predictors. This feature could become especially useful in future work aiming to extend the MAGICC-MESMER chain to emulate additional variables such as precipitation.

To increase the accessibility of our emulation chain and thus the use of such targeted climate information, MESMER (https://github.com/MESMER-group/mesmer) and the MAGICC-MESMER coupler (https://github.com/MESMER-group/mesmer
-openscmrunner) are already publicly available under GNU General Public License version 3 (GPL-3.0), while MAGICC is in the process of becoming open source. In the meantime, MAGICC output for various emission scenarios has been published as part of RCMIP Phase 1 (Nicholls et al., 2020) and 2 (Nicholls et al., 2021c) and could be translated to the local scale with the currently publicly available parts of the MAGICC-MESMER chain.

*Code and data availability.*  MESMER (https://github.com/MESMER-group/mesmer) and the MAGICC-MESMER coupler MESMER-
OPENSCMRUNNER (https://github.com/MESMER-group/mesmer-openscmrunner) are publicly available on GitHub and the versions employed in this study are archived on Zenodo (Beusch et al., 2021; Nicholls et al., 2021a). The MESMER documentation is hosted on



**Table A1.** List of the 25 employed CMIP6 models and the modeling groups providing them.

| Model | Modeling Center (or Group) |
|---|---|
| ACCESS-CM2 | Commonwealth Scientific and Industrial Research Organisation, Australia; Australian Research Council Centre of Excellence for Climate System Science, Australia |
| ACCESS-ESM1-5 | Commonwealth Scientific and Industrial Research Organisation, Australia |
| AWI-CM-1-1-MR | Alfred Wegener Institute, Helmholtz Centre for Polar and Marine Research, Germany |
| CanESM5 | Canadian Centre for Climate Modelling and Analysis, Environment and Climate Change, Canada |
| CESM2 | National Center for Atmospheric Research, Climate and Global Dynamics Laboratory, USA |
| CESM2-WACCM | National Center for Atmospheric Research, Climate and Global Dynamics Laboratory, USA |
| CMCC-CM2-SR5 | Fondazione Centro Euro-Mediterraneo sui Cambiamenti Climatici, Italy |
| CNRM-CM6-1 | Centre National de Recherches Météorologiques, France / Centre Européen de Recherche et Formation Avancée en Calcul Scientifique, France |
| CNRM-CM6-1-HR | Centre National de Recherches Météorologiques, France / Centre Européen de Recherche et Formation Avancée en Calcul Scientifique, France |
| CNRM-ESM2-1 | Centre National de Recherches Météorologiques, France / Centre Européen de Recherche et Formation Avancée en Calcul Scientifique, France |
| E3SM-1-1 | E3SM-Project and RUBISCO National Laboratories, USA |
| FGOALS-f3-L | Chinese Academy of Sciences, China |
| FGOALS-g3 | Chinese Academy of Sciences, China |
| FIO-ESM-2-0 | First Institute of Oceanography, Ministry of Natural Resources, China; Qingdao National Laboratory for Marine Science and Technology, China |
| HadGEM3-GC31-LL | Met Office Hadley Centre, UK; Natural Environment Research Council, UK |
| HadGEM3-GC31-MM | Met Office Hadley Centre, UK |
| IPSL-CM6A-LR | Institut Pierre Simon Laplace, France |
| MCM-UA-1-0 | Department of Geosciences, University of Arizona, Tucson, USA |
| MPI-ESM1-2-HR | Max Planck Institute for Meteorology, Germany; Deutscher Wetterdienst, Germany; Deutsches Klimarechenzentrum, Germany |
| MPI-ESM1-2-LR | Max Planck Institute for Meteorology, Germany; Alfred Wegener Institute, Helmholtz Centre for Polar and Marine Research, Germany |
| MRI-ESM2-0 | Meteorological Research Institute, Japan |
| NESM3 | Nanjing University of Information Science and Technology, China |
| NorESM2-LM | NorESM Climate modeling Consortium, Norway |
| NorESM2-MM | NorESM Climate modeling Consortium, Norway |
| UKESM1-0-LL | Met Office Hadley Centre, UK; Natural Environment Research Council, UK; National Institute of Meteorological Sciences/Korea Meteorological Administration, Climate Research Division, Republic of Korea; National Institute of Water and Atmospheric Research, New Zealand |

Read the Docs (https://mesmer-emulator.readthedocs.io/en/stable/). The scripts to create the figures in this paper can be found on GitHub (https://github.com/MESMER-group/Beusch_et_al_GMD_2021_MAGICC-MESMER_coupling) and are additionally archived on Zenodo (Beusch, 2021). The MAGICC data used in this study is available via Nicholls et al. (2020) and Nicholls et al. (2021c). The stratospheric

aerosol optical depth data employed during MESMER training are provided by NASA and available at https://data.giss.nasa.gov/modelforc e/strataer/. The CMIP6 data are available from the public CMIP archive at https://esgf-node.llnl.gov/projects/esgf-llnl/.

## Appendix A

*Author contributions.* L.B. initiated the study, wrote the initial implementation of MESMER, carried out all analyses, and wrote the first draft of the paper. Z.N. provided all the MAGICC data, supported the use of the data, wrote the initial implementation of the MAGICC-
MESMER coupling, and wrote the first draft of the MAGICC description section. L.B., M.H., and Z.N. currently co-develop the MESMER





and MAGICC-MESMER code bases. All authors designed the study together, discussed the results together, and contributed to improving the manuscript.

*Competing interests.*  The authors declare that they have no conflict of interest.

*Acknowledgements.*  We acknowledge the World Climate Research Program's Working Group on Coupled Modelling, which is responsible
for the Coupled Model Intercomparison Project (CMIP), and we thank the climate modelling groups (listed in Appendix Table A1) for producing and making available their model output. Furthermore, we are indebted to Urs Beyerle, Lukas Brunner, and Ruth Lorenz for pre-processing the CMIP6 data. We additionally thank Shruti Nath for her comments on an earlier version of this manuscript. L.B. acknowledges support from SNF grant P1EZP2_195662. S.I.S. acknowledges partial support from the ERC Proof-of-Concept Grant 964013 "MESMER-X".



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
