# Peer review of "From emission scenarios to spatially resolved projections with a chain of computationally efficient emulators: MAGICC (v7.5.1) – MESMER (v0.8.3) coupling"

_Geoscientific Model Development, 2021_

## Author Comment (AC1)

**Answer to reviewers:**

We thank Christopher Smith and Ben Sanderson for their constructive and valuable feedback on our manuscript. In the following, we provide point-by-point answers in blue to their reviews shown in black.

**Answer to Christopher Smith:**

In this paper the authors couple a global emulator (MAGICC) with a regional ESM emulator (MESMER) to create simulated fields of regional surface temperature anomalies from pre-industrial under different future climate scenarios. The combined emulator can be run either to simulate the response to particular CMIP6 models, or in an "observationally constrained" manner in which the global mean response ensemble from MAGICC has been pre-selected to be consistent with observed temperature and global mean uptake. This framework will be very useful to provide regional projections of climate change to future scenarios. Most importantly, and possibly a point that the authors undersell, is that the regional climate projections from any scenario can be produced, not just those that were run by ESMs (e.g. the SSPs). This would allow specific scientific questions to be answered such as the regional responses to different pre-defined levels of global mean warming, forcing, or total carbon budgets, and regional climate change commitments. The cheap computational framework, allowing millions of ensemble members to be run (memory is suggested as a limiting factor, but not processing speed), allows for statistically robust projections of climate risk in different regions and can aid adaptation planning. The modular, open-source framework appears to be flexible and extensible: while only annual mean regional temperature anomalies are included at the moment, the opportuntity to include precipitation and other climate variables (provided robust predictors are found), or higher simulated time resolution, should be possible without redesigning the whole framework.

This paper does something important and useful, and does it well, so my comments are limited to being quite minor.

Thank you for your positive evaluation of our study. In the revised mansucript, we will include a few additional remarks about our emulator chain being able to produce regional emulations for any emission scenario, not just the ones already run by ESMs.

lines 19-20: I assume IPCC 2021 also says this - though given the increased roles of emulators in the Sixth Assessment, perhaps ESMs are not our primary tools any more!

In the revised manuscript, we will only refer to regional climate change in this sentence, to avoid underselling the role global mean emulators have played in AR6.

figure 2: It took me a few minutes to fully decipher what the right side was showing here. Faint horizontal lines that separate the four rows would make it clearer that the spatial plots correspond to the same rows as the time series plots. In the first map in row (b) I think is taken that the pink, purple and orange lines are very similar and the forced GSAT plot from the blue line is a bit warmer. I would also take from this that in (b) and (c), dividing any of the four scenarios by any of the others would give a constant pattern scaling ratio of local to global GSAT that is the same everywhere.

Many thanks for letting us know that the readibility of Fig. 2 is not optimal yet. We will re-work the figure for the revised submission along the lines suggested while aiming to increase the overall accessibility of the content.

lines 139-140: more for my interest, but what regression coefficient did you determine for the stratospheric aerosol optical depth? Also, if you wanted to be totally CMIP6-consistent you could use the CMIP6 SAOD time series (ftp://iacftp.ethz.ch/pub_read/luo/CMIP6/CMIP_1850_2014_extinction_550nm_strat_only_v3.nc) rather than NASA-GISS one.

There are 25 different ESMs considered in this study and thus, we obtain 25 different SAOD regression coefficients. The calibrated coefficients vary between -2.86 and -0.46 with a median value of - 1.59 °C/SAOD unit. Note that for this user-oriented study, we choose to limit our focus on the emulations themeselves rather than on providing analyses for the calibrated parameters, since we expect the emulations to be more directly useful for a broader community. We will, therefore, not include any calibrated regression coefficient values in the main manuscript. Naturally, all of MESMER's calibrated parameters can be obtained by any interested reader with access to CMIP6 output by using the code we provide on GitHub.

Thank you for pointing us towards the CMIP6 SAOD time series data set. We will evaluate whether to retrain MESMER with this time series for the revised manuscript. However, since this CMIP6 SAOD data set requires additional pre-processing steps (it contains the SAOD values for different latitudes and altitudes, whereas our thus far employed time series is already available as a global average) and is more challenging to download (username and password are asked, whereas a direct download is possible for the thus far employed time series), we currently lean towards keeping our original data set for this study.

lines 193-194: of course, more predictors will reduce error. I was satisfied that the possibility of overfitting is addressed later on (lines 216-218). Still, I think the flow of the paper is a bit disjointed: we have a standard MESMER config (Beusch et al. 2020), then we introduce a relationship with more predictors and shows it performs slightly better (fig 3 and 4), but actually we don't use it for the main MESMER-MAGICC coupling in section 3.3 (explained in lines 249-255) that is the main subject of the paper. Perhaps a reordering to put the "additional predictors" section later on could be explored.

After careful consideration, we have decided to leave the manuscript structure as it currently is. While we understand you concerns about the a bit disjointed flow, we fear that reordering would lead to a larger thematic disjoint than the one our manuscript already encompasses. With the current ordering we discuss MAGICC, MESMER, and MAGICC-MESMER individually. Reordering the manuscript would force us to jump back and forth between the three topics.

line 247: I'm in general agreement with the authors' opinion on the implausibility of SSP5-8.5 but I think Hausfather & Peters 2020 gets abused a bit - particularly as we're running everything concentration driven here and high-end climate responses to a SSP3-7.0 (or even SSP4-6.0) like emissions pathway can't be ruled out.

In the revised manuscript, we will rephrase this sentence to read: "Given that SSP5-8.5 is designed to represent an unlikely high-risk future (Hausfather and Peters, 2020), one could justify excluding it from the local variability training to further improve the local variability emulations for the other scenarios." If

we were to exclude SS5-8.5 from training, this would not affect our ability to reproduce high-end climate responses in SS3-7.0 or SSP4-6.0. As long as ESM runs with high-end climate responses are available for training, MESMER can be calibrated to reproduce their behaviour.

line 268: MESMER (typo)

Unfortunately we cannot find the typo you refer to here. Before submitting our revised manuscript, we will carefully check for typos again and hopefully find the remaining ones.

lines 285-286: "however, most CMIP6 ESMs perform in an observationally-consistent manner in most regions (Beusch et al. 2020b)." For the ignorant such as me, you might want to briefly explain the discrepancy between poorly performing global GSAT and well-performing regional GSAT in ESMs. I don't know if it is true, or covered in Beusch et al. 2020b, but is it because the regions are mostly land regions and cover 30% of the surface, so much of the poorly performing regions with respect to observations are over the ocean?

Thank you for pointing out that our original description leaves room for misunderstanding. We will rephrase this sentence in the revised manuscript. The high warming GSAT ESMs generally also exhibit high regional land warming. In Beusch et al 2020b, we did not evaluate the ESMs' performance in terms of regional land warming itself, but instead, we evaluated whether their scaling factor between global and regional warming is consistent with the scaling factor that can be derived from observations. I.e., the regionally-aggregated regression coefficients $\beta^{fr}_{m,s}$ of most ESMs were found to be consistent with the ones derived from observations in most regions.

line 301: minor style typo: 360,000

Will be changed in the revised manuscript.

lines 304-306 (remark only, no need to respond): I believe this sparse sampling would be sufficient actually. We did some pattern scaling with FaIR where the global delta T from FaIR was combined with one of 10 ESM simulatons chosen at random where the ESM performed well over the UK domain. Indeed, the full span of uncertatiny was well sampled.

Interesting to hear that your own analyses confirm our hypothesis that also a sparser sampling would be sufficient to adequately sample the full uncertainty range. In the revised manuscript, we will nevertheless continue to use the brute force sampling approach, to avoid introducing additional potential uncertainty sources, which would need to be further explained and quantified.

figure 5: it could be a PDF rendering issue, but it would look nice if the alpha (1 - transparency) value for the shaded regions was < 1 to see the overlapping regions between blue and orange.

Actually, these are not shaded regions but individual overlapping emulation realizations. When revising the manuscript, we will trial the use of lower alpha values and alike to increase the information content in the overlapping purple and orange regions.

line 311: SSP1-1.9

Thanks, this typo will be corrected in the revised manuscript.

section 4.2: is it worth saying where these 600 constrained ensemble members came from? Is it the MAGICC AR6 WG1 config, or is it from one of the RCMIP papers - Nicholls et al. 2020?

The 600 member constrained MAGICC ensemble comes from the RCMIP phase 2 paper. This is first stated in Sect. 3.1 MAGICC (v7.5.1) and further details are provided in Sect. 3.3.12 Globally-constrained probabilistic emulations.

lines 336-337: optinally, you could hammer this home by giving the mean and range of ECS of these 6 models, compared to the full CMIP6 ensemble and/or the AR6 assessed range.

In light of the scope of our mansucript and the target audience we envision for it, we prefer to remain qualitative in our statements here, rather than needing to additionally dive into the definition and meaning of ECS and its relationship with near-term warming projections.

line 338: "is clearly incompatible ... (Fig. 6)." Well, only for 2 out of the 6 models - the other 4 look reasonable to me

Yes, that is precisely what we meant with our sentence. We will rephrase this sentence in the revised manuscript to avoid risking to mislead our readers.

lines 358-360: Indeed, there was a whole unofficial MIP (PDRMIP) devoted to this behaviour in ESMs. Tom Richardson derived a global precipitation emulator based on emissions of different GHGs, aerosols and GSAT: https://doi.org/10.1175/JCLI-D-17-0240.1. He had a regional one somewhere too but don't think it ever made it into a publication. Definitely something to explore.

Interesting, thank you for making us aware of this unofficial MIP as well as of Richardson's global precipitation emulator. We will reference the associated studies in our revised discussion.

**Answer to Ben Sanderson:**

Beusch et al discuss an emulation framework for the prediction of climate change impacts as a function of emissions scenarios. The framework is structured around the coupling of the MAGICC global simple climate model, and the MESMER regional pattern scaling model.

This paper presents the overall framework, including the methodology for the calibrations of the model components. It discusses the evaluation of regional results (as assessed through the metric of annual mean temperature), and the comparison of scenario projections with MAGICC-MESMER as compared with simulations in the CMIP6 archive.

The paper is clear, and well written and provides a novel and useful tool for the impacts community. I have no major issues barring publication, just some minor revisions are required to discuss some of the structural limitations which are implicit in the approach, and some comments which might help motivate further study.

Thank you for your overall positive feedback. We would, however, advise against referring to MESMER as a pattern scaling model, since in climate science, the term pattern scaling is overwhelmingly used for approaches which solely emulate the mean forced response. MESMER goes beyond this by additionally emulating internal climate variability.

Minor Issues

1 -In the discussion of limitations and future developments, it should be noted that the model structure allows for no dependency of climate variability on warming level - though there are probably elements of internal variability which are themselves dependent on warming or forcing level (Zheng 2018, Pendergrass 2017, Dorr 2021). These limitations, and potential for future developments, should be discussed a little more.

This limitation of our approach is mentionned in Sect. 3.2.4 Evaluating both MESMER configurations (lines 242-250 in the originally submitted manuscript) and it has additionally been discussed in the original MESMER description paper (Beusch et al., ESD, 2020). Based on your comment, we will furthermore introduce a short remark on how non-stationarity in internal climate variability across warming levels could be addressed within the MESMER framework in the discussion. For example, the monthly MESMER version MESMER-M, which has been developed under the lead of Shruti Nath (Nath et al 2021, preprint: https://doi.org/10.5194/esd-2021-59), follows the MESMER framework but integrates a more complex algorithm to account for non-stationarity of monthly temperature variability in a changing climate.

2- Furthermore, the internal variability is represented as a pattern related to global mean temperature deviations from a forced trajectory, plus a random term with imposed regional correlations through kriging. It is unclear from the present study whether this approach adequately reproduces (in a stationary climate) the noise-covariance structure of the original ESM which is being emulated, and the tests employed here - which focus on point-level errors - do not assess the skill of the emulator in producing realistic modes of natural variability. Though it's way beyond the scope here, and not necessary for this model overview, a future review could consider the relative performance of noise representation in the current scheme and other approaches (e.g. Perkins 2020, Alexeeff 2018, Holden 2010)

Yes, in the current study, we only evaluate grid-cell-level performance of our emulator. However, the questions you raise have been addressed in great detail in our original MESMER description paper (Beusch et al, 2020: https://doi.org/10.5194/esd-11-139-2020). While we restricted the quantitative assessment of the emulated variability to MESMER itself in that study, we discussed multiple alternative approaches qualitatively (including Alexeeff et al 2018 and Holden et al 2010). As a summary of the results presented in Beusch et al., 2020: the currently implemented MESMER approach for variability emulation reproduces local to regional cross-correlations well, but increasingly dampens covariances between grid points with increasing distance between them, due to a trade off between robust parameter estimations and available training data in the residual local variability module. Hence, MESMER's variability is by design increasingly underdispersive for larger regional averages. Only for global land averages, MESMER produces fully reliable emulations again, since variability at that scale is captured through the aggregated linear response of the individual grid cells to the global variability predictor. We actually already included a very short side remark to this in lines 141 – 142 (Sect. 3.2.1 Default configuration – Local temperature anomalies as a function of global temperature anomalies) of the originally submitted manuscript. In the revised manuscript we will expand on this side remark and additionally add a sentence about it in the discussion.

3 - the use of a lowess time filter to distinguish forced and variable components may exclude low frequency elements of natural variability which would ultimately be excluded from the model. The authors could test this in cases where large initial condition ensembles are available by combining ensemble members to produce an improved estimate of the underlying forced signal, smoothing (or not, if the ensemble is very large) and using the residual to estimate the noise component of the timeseries.

We agree with your evaluation that the simple LOWESS time filter currently implemented to extract the global mean forced response is in danger of excluding low frequency elements from the global variability and assigning them to the global forced trend instead, if the ESM at hand exhibits a pronounced low frequency behaviour (with decadal and longer variations in the global mean temprature variability around the global forced warming trend) and has published only a very limited number of initial condition ensemble members. The more initial condition ensemble members (sampling different phases of the low frequency variability at the same point in time) are included in the training, the less problematic our approach becomes for these pronounced low frequency variability ESMs. This is because the global temperature trend module first pools all available initial condition enesmble members and averages across them before using the LOWESS smoother on the remaining single time series. Hence, the more ensemble members are available, the more low frequency variations can already be removed in the initial averaging step before the LOWESS smoother is applied. Through your comment we realized that a remark on the preporcessing step of the data for the LOWESS smoother is missing from the original manuscript. We will include a sentence about it in Sect. 3.2.1 Default configuration – Local temperature anomalies as a function of global temperature anomalies in the revised manuscript.

The suggestion to consider very large initial condition ensembles is interesting indeed, especially as test beds for finding ways to improve our ability to separate global forced warming from natural variability for ESMs with pronounced low frequency variability and limited numbers of available simulations. Note, however, that exploring this option is beyond the scope of this study, which rather focuses on bringing two previously published approaches (MAGICC and MESMER) for computationally efficient Earth system modelling together.

4 - The introduction of the additional predictors (a quadratic dependency of regional temperatures and a term dependent on global ocean heat uptake) are interesting extensions to the model. My concern is whether there is sufficient data to unambiguously fit these additional degrees of freedom, and whether there is spatial coherence in the relative role of the non-linear term and the ocean heat uptake term over the gridded field. As the authors move towards a larger number of predictors, they might want to consider an EOF prefilter for spatial fields - allowing the model parameters to be fitted in a lower dimensional space which enforces covariance structure. This framework might also aid ultimately in the probabilistic calibration of the MESMER component.

As highlighted by the sensitivity experiments in lines 204 – 220 (Sect 3.2.3 Evaluating both MESMER configurations) of the originally submitted manuscript, we are confident that using the historical time period plus a future high emission (here SSP5-8.5) as well as a strong future mitigation scenario (here SSP1-2.6) are sufficient to unambigiously fit the additional degrees of freedom introduced by the additional predictors we employ in this study. The emulators calibrated on this subset of scenarios emulate the remaining scenarios similarly well as if all future scenarios had been included during training, indicating that they did not overfit on their limited training data (Fig. S7). If a single future emission scenario is used for the training however, strong overfitting to that scenario type occurs with poor performance in other scenarios (especially in the ones that differ the most from the training scenario). Generally, spatially rather coherent calibrated parameter maps are obtained for each local forced response regression parameter (not shown). This is expected due to the underlying nature of temperature fields which generally vary smoothly in space and in which individual grid point time series are often strongly correlated with the time series of neighbouring grid points. Hence, when fitting the regression coefficients on individual grid cells, neighbouring grid cells tend to have similar values assigned to them.

Naturally, there are limits to this statement with some regions showing more noisy parameter maps in some ESMs. Since the main focus of the manuscript is on MESMER's default configuration and not on the additional predictors configuration, and on the emulations, not on the calibrated parameters, we prefer to not add any additional plots concerning this topic to our manuscript. We agree that once we move to a larger number of predictors, more thought needs to put into constraining calibrated parameters in a physically meaningful way, without ending up with a set of highly correlated predictors and noisy calibrated parameter fields. We will add a few sentences about this topic to Sect. 5.1 Going beyond global mean temperature as a predictor for the regional scale in the revised manuscript. An EOF prefilter for the spatial fields is an interesting idea for systematically reducing the dimensionality of the statistical emulation challenge. We will consider this for future developments of the MESMER emulator.

5 - The ocean heat uptake term is a very useful extension to MESMER for representing temperatures in deep mitigation scenarios. In future versions, the authors might find it useful to partition the heat uptake by depth, to represent the pattern effects of heat stored in the oceanic mixed layer as distinct from the deep ocean.

Thank you for this suggestion. We will add a short mention of it in Sect. 5.1 Going beyond global mean temperature as a predictor for the regional scale.

6 - There a brief discussion of the role of non-GHG forcers in the current study, and how this might be incorporated in the future. A brief note on how the simplified current framework might therefore introduce bias would be useful. i.e. to what degree are aerosol/ghg pathway co-dependencies 'baked into' the MESMER configration, and is this evident by looking at scenario outliers like SSP3-RCP7?

As you rightly point out, some correlation / co-dependence of the GHG pathways with aerosols is 'baked in' in the current approach. E.g., since no aerosol predictors are included in either MESMER configuration, local temperature effects of regional aerosol emissions (e.g., documented by Lund et al 2020: https://doi.org/10.5194/esd-11-977-2020 or Persad & Caldeira 2018: https://doi.org/10.1038/s41467-018-05838-6) cannot be captured by MESMER's emulations for the different SSP scenarios. We will add a remark about this caveat in the revised discussion. Nevertheless, based on Figures 3, S6, and S7 and the overall encouraging emulation results on all SSP emission scenarios (including outlier scenarios such as SSP3-7.0 & SSP5-3.4-over) even when training solely on Historical + SSP1-2.6 + SSP5-8.5, we are convinced that our simple approach is largely sufficient to emulate annual mean temperatures in CMIP6-like future emission scenarios in most regions despite not considering aerosol time series.

References:

Zheng, Xiao-Tong, Chang Hui, and Sang-Wook Yeh. "Response of ENSO amplitude to global warming in CESM large ensemble: uncertainty due to internal variability." Climate Dynamics 50.11 (2018): 4019-4035.

Pendergrass, A. G., Knutti, R., Lehner, F., Deser, C., & Sanderson, B. M. (2017). Precipitation variability increases in a warmer climate. Scientific reports, 7(1), 1-9.

Dörr, J., Årthun, M., Eldevik, T. and Madonna, E., 2021. Mechanisms of regional winter sea-ice variability in a warming Arctic. Journal of Climate, 34(21), pp.8635-8653.

Perkins, W. Andre, and Greg Hakim. "Linear inverse modeling for coupled atmosphere-ocean ensemble climate prediction." Journal of Advances in Modeling Earth Systems 12.1 (2020): e2019MS001778.

Alexeeff, Stacey E., et al. "Emulating mean patterns and variability of temperature across and within scenarios in anthropogenic climate change experiments." Climatic Change 146.3 (2018): 319-333.

Holden, P. B., and N. R. Edwards. "Dimensionally reduced emulation of an AOGCM for application to integrated assessment modelling." Geophysical Research Letters 37.21 (2010).

---

## Author Response (AR1)

Dr. Lea Beusch
Institute for Atmospheric and
Climate Science ETH Zurich
Universitaetstrasse 16
CH-8092 Zurich
E-mail: lea.beusch@env.ethz.ch

[Figure]

ETH
Eidgenössische Technische Hochschule Zürich
Swiss Federal Institute of Technology Zurich

Geoscientific Model Development Editorial Board

Minusio, 3 January 2022

**From emission scenarios to spatially resolved projections with a chain of computationally efficient emulators: MAGICC (v7.5.1) – MESMER (v0.8.3) coupling**

Dear Dr. O'Connor,

Please find enclosed the revised manuscript, the revised supplement, and a tracked changes version of the manuscript. For the answer to the reviewers, we refer to the point-by-point answers we posted in the interactive discussion.

During the revisions, we introduced the following main changes:

1. The discussion and parts of the methods were extended to address the aspects raised by the reviewers (for details, please refer to the point-by-point answer we posted in the interactive discussion).

2. MESMER's current version (v0.8.3) was calibrated using the new stratospheric aerosol optical depth (SAOD) time series employed by the sixth assessment report of the Intergovernmental Panel on Climate Change (IPCC) and all Earth System Model (ESM) runs available in our archive for the 25 ESMs considered in this study to create a new set of emulations. While the new emulations exhibit minor quantitative differences compared to the original emulations, no qualitative findings were affected by these changes. For the analyses carried out in this paper, MESMER v0.8.1 and v0.8.3 lead to identical results. Hence, the small differences in the emulations, which are visible for some ESMs (see e.g., Figs. 3 and 4) can be attributed to the new SAOD time series as well as to the available ESM runs (e.g., an additional scenario became available in our archive for NorESM2-LM).

3. Figure 2 and its caption have been reworked to increase readability.

4. A small bug in the code that derives the data for plotting in the figures comparing different MESMER configurations was removed (the emulation error was mistakenly depicted as zero in some cases in the originally submitted study). Furthermore, the labels in these figures as well the figure captions were rewritten to increase the accuracy of their description.

Additionally, minor changes have been introduced throughout the text to improve text flow and reduce typos. These can all be found within the tracked changes version of the manuscript.

We are confident that the revisions increased the quality of our study.

Yours sincerely,

Lea Beusch
(on behalf of all co-authors)

---

## Author Response (AR2)

Dr. Lea Beusch
Institute for Atmospheric and
Climate Science ETH Zurich
Universitaetstrasse 16
CH-8092 Zurich
E-mail: lea.beusch@env.ethz.ch

[Figure]

Geoscientific Model Development Editorial Board

Minusio, 2 February 2022

**Technical corrections for study: From emission scenarios to spatially resolved projections with a chain of computationally efficient emulators: MAGICC (v7.5.1) – MESMER (v0.8.3) coupling**

Dear Fiona,

Thank you for giving us the opportunity to still include technical corrections before the manuscript goes into production. Please find below a full list of the changes we introduced in the article and the associated line numbers in the marked-up manuscript we submitted as part of the revisions:

- We made an update to Fig. 1, which is visually hardly detectable since only very minor changes were introduced in the plots. In the revised manuscript, the temperature realizations in Fig. 1 were unfortunately still based on the emulations we created for the initial submission. In the updated figure, they are based on the emulations we generated for the revised manuscript.
- We corrected a typo in the regionmask version in line 90.
- We replaced "selected" with "select" in line 217.
- We removed an unnecessary pointer towards Fig. S6 from line 226.
- We removed an unnecessary comma in line 240.
- We changed "true" to "actual" in line 251.
- We corrected "dev/iation" typo in the caption of Fig. 4.
- We removed "Nevertheless" in line 267.
- Thanks to your feedback, we were able to finally correct the "MESER" typo in line 275.
- We changed "v7.1.0.-beta" to "v7.1.0-beta" in line 282 to increase consistency in the version notation.
- We changed the order of the references in lines 301-302 to follow the same rules as everywhere else.
- We improved the text flow in the sentence in line 311-313.
- We added a missing hyphen to "short-lived" in line 377.
- We replaced "regional emulators'" with "spatially resolved emulators'" in line 453, since we do not use the term "regional emulator" anywhere else in the study.
- We changed "sets" to "set" because using a plural does not make sense in line 458.
- We changed "supplement" to "Supplement" throughout text.
- We harmonized the writing of "emissions-driven" throughout the text.
- We visited the websites in the References again and updated the last accessed dates accordingly.
- In the Supplement, in the caption of Fig. S7, we changed "Figure" to "Fig.".

All the best,
Lea
(on behalf of all co-authors)